# Phytosterols Are Involved in Sclareol-Induced Chlorophyll Reductions in *Arabidopsis*

**DOI:** 10.3390/plants12061282

**Published:** 2023-03-11

**Authors:** Asma Ben Hmidene, Hiroshi Ono, Shigemi Seo

**Affiliations:** 1Crop Disease Research Group, Division of Plant Molecular Regulation Research, Institute of Agrobiological Sciences, National Agriculture and Food Research Organization, 2-1-2 Kannondai, Tsukuba 305-8518, Ibaraki, Japan; 2Bioactive Chemical Analysis Unit, Research Center for Advanced Analysis, National Agriculture and Food Research Organization, 2-1-12 Kannondai, Tsukuba 305-8642, Ibaraki, Japan

**Keywords:** sclareol, *Arabidopsis*, chlorosis, chlorophyll reduction, phytosterols, campesterol, stigmasterol

## Abstract

Sclareol, a diterpene, has a wide range of physiological effects on plants, such as antimicrobial activity; disease resistance against pathogens; and the expression of genes encoding proteins involved in metabolism, transport, and phytohormone biosynthesis and signaling. Exogenous sclareol reduces the content of chlorophyll in *Arabidopsis* leaves. However, the endogenous compounds responsible for sclareol-induced chlorophyll reduction remain unknown. The phytosterols campesterol and stigmasterol were identified as compounds that reduce the content of chlorophyll in sclareol-treated *Arabidopsis* plants. The exogenous application of campesterol or stigmasterol dose-dependently reduced the content of chlorophyll in *Arabidopsis* leaves. Exogenously-applied sclareol enhanced the endogenous contents of campesterol and stigmasterol and the accumulation of transcripts for phytosterol biosynthetic genes. These results suggest that the phytosterols campesterol and stigmasterol, the production of which is enhanced in response to sclareol, contribute to reductions in chlorophyll content in *Arabidopsis* leaves.

## 1. Introduction

Disease control strategies that utilize a plant’s natural resistance to pathogens are attractive to researchers because of the potential to reduce environmental impact. Some of the main materials used in disease control techniques are disease-resistance-inducing compounds, characterized by the capability to induce resistance to a broad range of pathogens without targeted antimicrobial activity. Many natural and synthetic compounds have been identified as disease-resistance-inducing compounds [1,2,3,4], among which the diterpene sclareol is attractive due to its physiological activities.

Pharmacological studies show that sclareol exhibits antimicrobial activity against plant pathogenic bacteria and fungi [5,6,7,8,9]. The exogenous application of sclareol induces the expression of genes encoding proteins involved in disease resistance, metabolism, transport, and phytohormone biosynthesis and signaling in *Nicotiana* spp. and *Arabidopsis* (*Arabidopsis thaliana*) [10,11,12]. Sclareol also induces resistance to a bacterial pathogen and plant-parasitic nematode in tobacco (*Nicotiana tabacum*), tomato (*Solanum lycopersicum*), and *Arabidopsis* plants without antibacterial or nematicidal activity [13,14]. While sclareol is produced in only a limited number of plant species, such as clary sage (*Salvia sclarea*) and *Nicotiana* species [15,16,17], it has been suggested that an ethylene signaling pathway mediates sclareol-induced disease resistance not only in tobacco, but also in *Arabidopsis* [13,14], a plant that does not produce sclareol. These findings indicate that a mechanism for sclareol-responsive signal transduction exists in plants.

Our research group recently reported that when exogenous sclareol was applied to *Arabidopsis* plants, the leaves developed chlorosis-like symptoms that were accompanied by a reduction in chlorophyll content [18]. However, whether such symptoms are related to sclareol-induced disease resistance remains unclear. As the first step to clarify the physiological meaning of sclareol-induced chlorosis-like symptoms, we tested the assumption that in response to exogenously-applied sclareol, *Arabidopsis* plants produce endogenous substances that decrease the content of chlorophyll. Herein, we identified and isolated phytosterols as chlorophyll-content-reducing substances.

## 2. Results and Discussion

### 2.1. Isolation and Identification of Chlorophyll-Content-Reducing Substances

First, we examined whether chlorophyll-content-reducing substances could be extracted with organic solvents. Because our preliminary experiments showed that in response to exogenous sclareol, four-week-old *Arabidopsis* plants exhibited an increase in chlorophyll reduction compared to eight-week-old plants, the growth stage used in our previous study [18], four-week-old plants were used in this study. We compared two common solvents, acetone and methanol, for extraction. When an acetone extract was applied to *Arabidopsis* plants, a significant reduction in chlorophyll was reproducibly observed. Such a reduction was not observed with the methanol extraction, although the cause is unknown. Therefore, we proceeded with an acetone extraction system in further experiments. To confirm whether chlorophyll-content-reducing substances exist in sclareol-untreated plants, we treated plants with 0.1% methanol—the solvent used to dissolve sclareol—which we characterized as the “mock treatment”. These samples were also subjected to the acetone extraction system in the same way. Chlorophyll content was lower in leaves treated with the acetone extract from sclareol-treated plants than in leaves treated with the acetone extract from the mock treatment (Appendix A). As a negative control, we also measured chlorophyll content in plants that were treated with only 0.1% methanol and not the acetone extract from sclareol-treated or mock-treated plants. There was no difference in chlorophyll content between those in 0.1% methanol and the mock treatment. These results suggest that endogenous substances that are involved in sclareol-induced chlorophyll reduction are acetone-extractable. Hereafter, we used a 0.1% methanol treatment as the negative control.

To purify and isolate acetone-extractable substances, we performed a large-scale extraction with three different solvents: ethyl acetate, hexane, and water. The content of chlorophyll was lower in all other fractions than in 0.1% methanol (Figure 1).

The hexane-soluble fraction exhibited slightly higher activity for reducing the content of chlorophyll than the ethyl acetate-soluble fraction. Therefore, we focused on the hexane-soluble fraction for further purification. The hexane-soluble fraction was fractionated by chromatography on a silica gel column followed by a C_18_-based solid-phase extraction (SPE) column. The active fractions obtained by SPE were subjected to preparative thin-layer chromatography (TLC) or silica gel column chromatography (Appendix A). Activity to reduce the content of chlorophyll was detected in two fractions: compound one was obtained from TLC, and compound two from silica gel column chromatography. These two active fractions were collected and purified. Resonances in ^1^H- and ^13^C-nuclear magnetic resonance (NMR) spectra for compounds 1 and 2 were assigned to campesterol and stigmasterol, respectively (Figure 2). Campesterol and stigmasterol are phytosterols that belong to isoprenoids, the largest class of natural compounds which play essential roles in plant growth and development, such as seed germination, architecture, and reproduction [19].

### 2.2. Effects of Campesterol and Stigmasterol on Chlorophyll Reductions

To examine the effects of campesterol and stigmasterol on chlorophyll reduction, Col-0 plants were treated by immersing their roots in a solution containing various concentrations (50, 100, and 200 μM) of campesterol or stigmasterol for 48 h, after which the chlorophyll contents of the treated plant leaves were measured. The effects of campesterol on chlorophyll reductions were detected at 100 μM or higher (Figure 3). Stigmasterol induced dose-dependent reductions in chlorophyll content. These results suggest that exogenously applied campesterol and stigmasterol reduced the chlorophyll content in *Arabidopsis* leaves.

### 2.3. Effects of Sclareol on Accumulation of Campesterol and Stigmasterol, and Expression of Phytosterol Biosynthetic Genes

A quantitative analysis showed that endogenous levels of campesterol and stigmasterol were approximately 11- and 15-fold higher, respectively, in sclareol-treated leaves than in methanol-treated leaves (Figure 4).

To examine whether sclareol-induced accumulation of campesterol and stigmasterol was associated with the induction of phytosterol biosynthetic genes, induction kinetics in sclareol-treated leaves were assessed for 3-hydroxy-3-methylglutaryl-CoA (*HMG-CoA*), reductase (*HMGR*), sterol 4α-methyl oxidase1 (*SMO1*), and sterol C-24 reductase (*DWF1). HMGR* catalyzes the conversion of *HMG-CoA* to mevalonate, a precursor of phytosterols [20]. *SMO1* and *DWF1* are involved in the biosynthesis of phytosterols, including campesterol and stigmasterol [21,22,23]. We also used *AtPDR12*, a sclareol-responsive gene [10,13], as a positive control to confirm the functionality of the experimental system. The treatment with sclareol enhanced transcripts for *AtHMGR*, *AtSMO1-2*, and *AtDWF1* in Col-0 leaves (Figure 5). Sclareol-induced accumulation of *AtPDR1*2 transcripts was also observed.

To examine whether phytosterols are involved in sclareol-mediated reduction of chlorophyll, the effect of lovastatin, an inhibitor of *HMGR*, on chlorophyll reduction in the presence of sclareol was tested. When lovastatin was only applied to plants, an increase in chlorophyll content was observed (Figure 6; MeOH vs. lovastatin), which is consistent with previous findings showing that lovastatin increases chlorophyll content when applied to *Arabidopsis* seedlings [24]. Sclareol-induced reduction in chlorophyll content was partially restored by lovastatin (Figure 6; sclareol vs. lovastatin + sclareol).

Collectively, these results show that the production of stigmasterol and campesterol in *Arabidopsis* leaves is enhanced by the up-regulated expression of phytosterol biosynthetic genes, such as *AtHMGR*, *AtSMO1-2*, and *AtDWF1*, in response to sclareol, suggesting that accumulated campesterol and stigmasterol contribute to a reduction in chlorophyll content. These results are consistent with previous findings showing that *Arabidopsis* mutants with mutations in phytosterol homeostasis genes had elevated levels of phytosterols, including campesterol-related and stigmasterol-related compounds, and showed early senescence [25,26].

During solvent–solvent fractionation after the acetone extraction of sclareol-treated plants, we detected chlorophyll-content-reducing activity in ethyl acetate- and water-soluble fractions, suggesting that substances other than campesterol and stigmasterol are also involved in chlorophyll reductions. To identify these active substances, future studies are warranted.

## 3. Materials and Methods

### 3.1. Plant Materials

*Arabidopsis* Columbia (Col-0) was used in this study. *Arabidopsis* plants were grown in a hydroponic system. Seeds were sown on half-strength Murashige and Skoog medium (Wako Pure Chemical, Osaka, Japan) supplemented with 0.8% agar and grown under a cycle of 8 h of light and 16 h of dark with a photon flux density of 120 μ mol m^−2^ s^−1^ at 22 °C. Ten-day-old seedlings were grown on a sheet of a polyethylene raft through holes (1 cm in diameter) floating over a 1500-fold diluted liquid fertilizer (Hyponex). The fertilizer was changed weekly, and their roots were soaked in the fertilizer under the light conditions described above. Unless otherwise stated, four-week-old plants were used.

### 3.2. Extraction, Fractionation, and Purification of Active Substances

#### 3.2.1. Small-Scale Extraction

The roots of four-week-old *Arabidopsis* Col-0 wild-type plants (10 g fresh weight) were immersed in a solution containing 100 μM sclareol for 48 h, the minimum concentration and sufficient time needed to decrease the content of chlorophyll in this plant species [18]. As the mock treatment, we also treated plants with 0.1% methanol in the same way. The treated plants were extracted by soaking the plants in five volumes (50 mL) of cold 80% (*v*/*v*) acetone or cold 80% (*v*/*v*) methanol at 4 °C for one week. After filtration, each extract was evaporated to dryness, and the remaining residue was dissolved in 20 mL of 0.1% methanol. The methanol solution was applied to Col-0 plants according to the procedure described in Section 3.3. The chlorophyll content of treated plant leaves was subsequently measured.

#### 3.2.2. Large-Scale Extraction

The plant extracts of sclareol-treated *Arabidopsis* plants (500 g fresh weight) were extracted by soaking the plants in five volumes of cold 80% (*v*/*v*) acetone at 4 °C for one week. After the removal of acetone under a vacuum using a rotary evaporator, the residual aqueous layer was partitioned with hexane followed by ethyl acetate. The hexane and ethyl acetate layers were regarded as the hexane- and ethyl acetate-soluble fractions, respectively. The remaining aqueous layer was regarded as the water-soluble fraction. The hexane layer (472 mg) was separated on a column (53 mm in diameter, 550 mm in length) of silica gel (Wakogel C-200, FUJIFILM Wako Pure Chemical, Osaka, Japan) eluted with increasingly higher concentrations of ethyl acetate in hexane. Activity for reducing chlorophyll content was detected in a fraction eluted with 30% (*v*/*v*) ethyl acetate in hexane and fractions eluted with 40 to 50% (*v*/*v*) ethyl acetate in hexane. The fraction eluted with 30% (*v*/*v*) ethyl acetate in hexane was collected and separated on a C_18_-based solid phase extraction (SPE) cartridge column (Waters) eluted with increasingly higher concentrations of methanol in water, starting with 0% methanol and ending with 100% methanol. A fraction eluted in 100% methanol was separated on a column of silica gel eluted with increasingly higher concentrations of chloroform in methanol to obtain two active fractions. One active fraction was further purified by preparative TLC (silica gel 60 F254, Merck, Rahway, NJ, USA) using 100% chloroform to yield compound 1 (1.1 mg), and another fraction was separated on a column of silica gel eluted with 10% (*v*/*v*) chloroform in methanol to yield compound 2 (2 mg).

All fractions obtained were evaporated, dissolved in methanol, diluted to appropriate concentrations with water, and used in the treatment of *Arabidopsis* plants to measure activity-reducing chlorophyll content.

### 3.3. Chemical Treatments

Sclareol was purchased from Sigma-Aldrich (St. Louis, MO, USA). Campesterol, stigmasterol, and lovastatin were purchased from Tokyo Chemical Industry (Tokyo, Japan). Sclareol, campesterol, stigmasterol, and lovastatin were dissolved in methanol and diluted in water to various concentrations. Methanol concentrations did not exceed 0.1% (*v*/*v*) in any experiment.

The procedure of chemical treatment, including fractions obtained from fractionation, was performed according to previous reports [16,18]. Briefly, plants were removed from the fertilizer, transferred to a glass Petri dish (9 cm in diameter) containing sterile distilled H_2_O, and preincubated for 24 h to diminish the influence of any stress wounds caused by the transfer. After discarding water with a pipette, plant roots were treated by gently adding a solution containing adequate concentrations of each fraction or compound for 48 h. The treated plants were harvested and used for chlorophyll measurements, gene expression analysis, and phytosterol quantification. Five to six plants were used for each treatment and regarded as one biological replicate.

For the lovastatin treatment, we used 100 μM lovastatin, 100 μM sclareol, and a mixture of 100 μM lovastatin and 100 μM sclareol in a 50:50 ratio.

### 3.4. Chlorophyll Measurement

Samples were extracted with cold 80% (*v*/*v*) acetone by grinding with a mortar and pestle on ice. The homogenate was centrifuged at 3000 rpm for 10 min. The supernatant was collected, and the precipitate was further extracted with cold 80% acetone. Supernatants were combined and used to measure absorbance at 646 and 663 nm with a spectrophotometer. Chlorophyll *a* and *b* contents were calculated according to the formula described by Porra et al. [27].

### 3.5. Structural Analyses of Active Compounds

#### 3.5.1. NMR

1D- and 2D-NMR spectra (1H, 13C, HSQC, HMBC, COSY, and NOESY) were performed using Bruker Avance-NMR (600 and 800 MHz) spectrometers. All compounds were recorded in the solvent deuterochloroform (CDCl_3_).

#### 3.5.2. TLC

TLC was performed on a precoated aluminum plate of silica gel 60 F_254_ (Merck) with the solvent system consisting of different ratios of chloroform and methanol. Chromatograms were visualized under ultraviolet (UV) light using a UV lamp for the localization of spots on chromatograms before and after spraying with the phosphomolybdic acid stain.

#### 3.5.3. Gas Chromatography–Mass Spectrometry

A gas chromatography–mass spectrometry (GC–MS) analysis was performed on a GC (7890A, Agilent, Santa Clara, CA, USA) system equipped with a mass selective detector (5975C, Agilent). Separation was performed on a capillary column (HP-5MS, length of 30 m, i.d. of 0.25 mm, thickness of 0.25 μm, Agilent) with He as the carrier gas at a flow rate of 1 mL min^−1^. The injection mode was splitless and the injection port temperature was 280 °C. The oven temperature was set at 50 °C, increased to 300 °C at 2 °C min^−1^, and held at 300 °C for 5 min. The mass spectrometer was operated in the electron impact mode at 70 eV and scan mode, scanning from 50 to 500 *m*/*z* at 6.35 scan^−1^. The compounds isolated in the GC–MS analysis were identified by matching their mass spectra to the reference spectra of the corresponding authentic standards.

#### 3.5.4. Silylation of Samples

Regarding the silylation of samples prior to the GC–MS analysis, 1 mg of compound was dissolved in 100 μL pyridine, and 100 μL *N*-Methyl-*N*-(trimethylsilyl)trifluoroacetamide (MSTFA; Sigma-Aldrich) was added to the pyridine solution. The mixture was heated to 60 °C for 3 h. After removal of the solvent under a nitrogen stream, the product obtained was dissolved in chloroform and injected into the GC–MS instrument.

#### 3.5.5. Compound 1

White powder; *Rf* 0.33 in CHCl_3_-MeOH (9:1); ^1^H-NMR (CDCl_3_, 800 MHz): 5.35 (m, 1H, H_6_), 3.52 (m, 1H, H_3_), 1.01 (s, 3H, H_19_), 0.85 (d, *J* = 6.4, 3H, H_26_), 0.80 (d, *J* = 7.2, 3H, H_27_), 0.77 (d, *J* = 7.2, 3H, H_28_), 0.68 (s, 3H, H_18_) and ^13^C-NMR data (CDCl_3_, 800 MHz): 140.8, 121.7, 71.8, 20.2, 19.4, 18.7, 18.2, 15.4; GC–MS *m*/*z* 472 [M + 72]^+^ (C_28_H_48_O).

#### 3.5.6. Compound 2

White powder; *Rf* 0.38 in CHCl_3_-MeOH (9:1); ^1^H-NMR (CDCl_3_, 800 MHz): 5.35 (m, 1H, H_6_), 5.15 (dd, *J* = 8.7, 5.3, 1H, H_23_), 5.01 (dd, *J* = 8.5, 5.2, 1H, H_22_), 3.52 (m, 1H, H_3_), 1.02 (d, *J* = 6.6, 3H, H_21_), 1.01 (s, 3H, H_19_), 0.91 (d, *J* = 8, 3H, H_21_), 0.84 (d, *J* = 6.5, 3H, H_26_), 0.81 (d, *J* = 7.4, 3H, H_27_), 0.79 (d, *J* = 6.6, 3H, H_29_), 0.69 (s, 3H, H_18_) and ^13^C-NMR data (CDCl_3_, 800 MHz): 140.8, 138.3, 129.3, 121.7, 71.8, 21.2, 21.1, 19.4, 18.9, 12.3, 12.1; GC–MS *m*/*z* 484 [M + 72]^+^ (C_29_H_48_O).

### 3.6. Quantification of Phytosterols

Extraction and derivatization using MSTFA of the samples were performed as previously described [28]. A GC–MS analysis of the sample was performed according to the conditions described above, except that selected ion monitoring was used for data acquisition at *m*/*z* 472 for campesterol and *m*/*z* 484 for stigmasterol.

### 3.7. Quantitative Real-Time PCR

The extraction of total RNA and quantitative real-time PCR using total RNA were performed in a two-step reaction using a SYBR Green kit (Bio-Rad, Hercules, CA, USA) in accordance with the procedure described by Fujimoto et al. [18]. Information on the primers used is shown in Appendix A. The expression levels of *Atactin2* were used to normalize those of the target genes.

### 3.8. Statistical Analyses

A one-way analysis of variance followed by Tukey’s test was used to evaluate the significance of differences within all groups. Student’s *t*-test was employed to compare the significance of differences in the mean of two samples. Dunnett’s test was used for Appendix A. These analyses were conducted using R version 2.13.1 (R Development Core Team, 2011).

## Figures and Tables

**Figure 1 plants-12-01282-f001:**
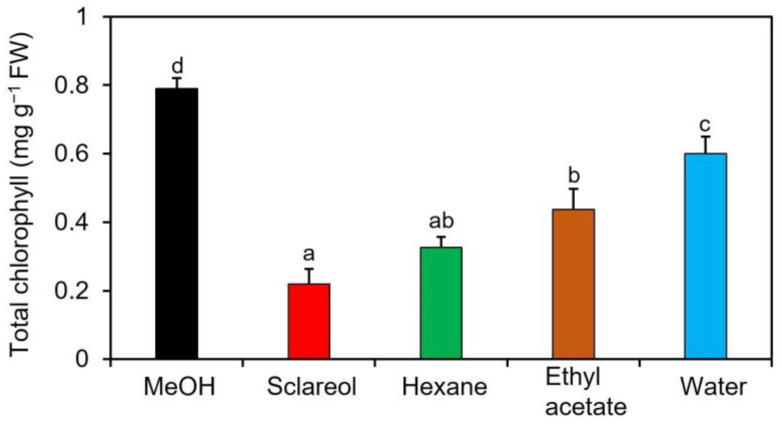
Effects of fractions obtained from an acetone extract of sclareol-treated plants on the accumulation of chlorophyll in *Arabidopsis* leaves. Sclareol-treated *Arabidopsis* plants were extracted with acetone. The extract was divided into hexane-, ethyl acetate-, and water-soluble fractions and applied to *Arabidopsis* plants by soaking their roots in a solution containing each fraction for 48 h. The leaves of the treated plants were collected and the contents of total chlorophyll, represented as the sum of chlorophyll *a* and *b*, were measured. As a negative control, 0.1% methanol (MeOH) was used, as previously described in this section. We used 100 μM sclareol as a positive control. Values are the means ± standard deviation of three biological replicates. Different letters indicate significant differences among treatments (Tukey’s test, *p* < 0.05).

**Figure 2 plants-12-01282-f002:**
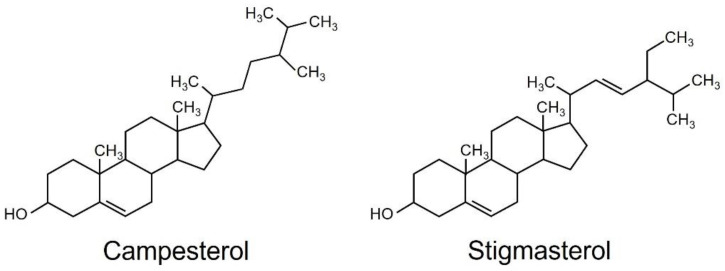
Structures of campesterol and stigmasterol.

**Figure 3 plants-12-01282-f003:**
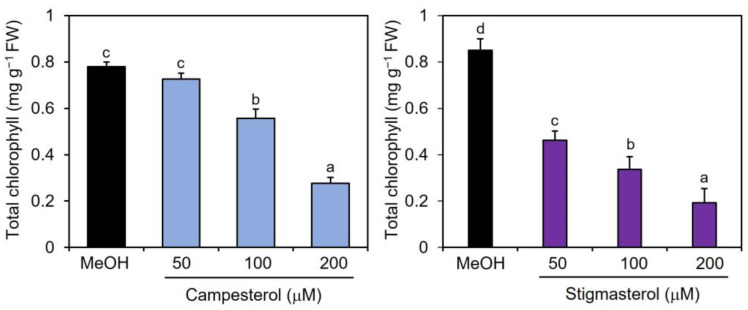
Effects of campesterol and stigmasterol on the accumulation of chlorophyll in *Arabidopsis* leaves. *Arabidopsis* plants were treated by soaking their roots in a solution containing various concentrations of campesterol or stigmasterol for 48 h. The leaves of the treated plants were collected and the total chlorophyll content, represented as the sum of chlorophyll *a* and b, was measured. As a negative control, 0.1% methanol (MeOH) was used. Values are the means ± standard deviation of three biological replicates. Different letters indicate significant differences among treatments (Tukey’s test, *p* < 0.05). The experiment was repeated three times with similar results.

**Figure 4 plants-12-01282-f004:**
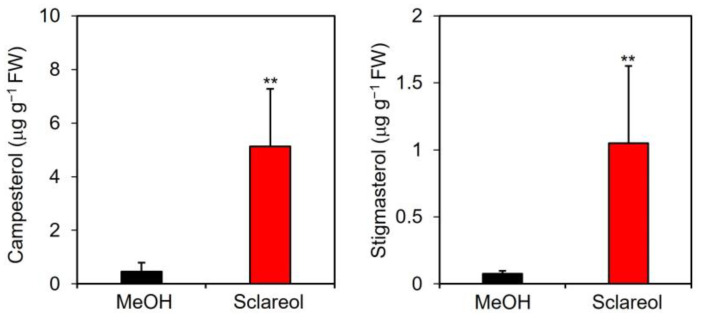
Effects of exogenous sclareol on the accumulation of campesterol and stigmasterol in *Arabidopsis* leaves. *Arabidopsis* plants were treated by soaking their roots in a solution containing 100 μM sclareol for 48 h. The leaves of the treated plants were collected and subjected to the measurement of campesterol and stigmasterol. As a control, 0.1% methanol (MeOH) was used. Values are the means ± standard deviation of three biological replicates. Asterisks denote significant differences from the 0.1% methanol sample (*t*-test, ** *p* < 0.01). The experiment was repeated three times with similar results.

**Figure 5 plants-12-01282-f005:**
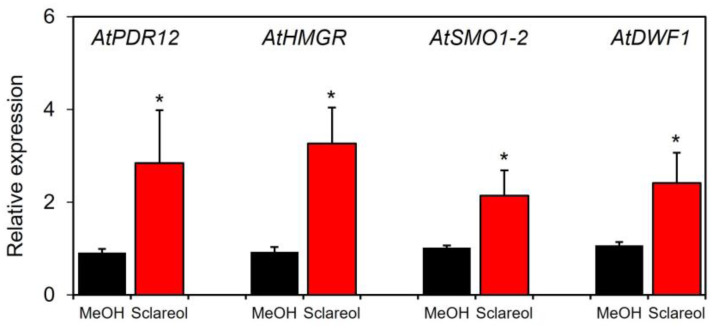
Effects of exogenous sclareol on the expression of phytosterol biosynthetic genes in *Arabidopsis* leaves. *Arabidopsis* plants were treated by soaking their roots in a solution containing 100 μM sclareol for 48 h. The leaves of the treated plants were collected and subjected to a quantitative real-time PCR analysis. As a control, 0.1% methanol (MeOH) was used. Values are the means ± standard deviation of three biological replicates. Asterisks denote significant differences from the negative control (0.1% methanol) sample (*t*-test, * *p* < 0.05). The experiment was repeated three times with similar results.

**Figure 6 plants-12-01282-f006:**
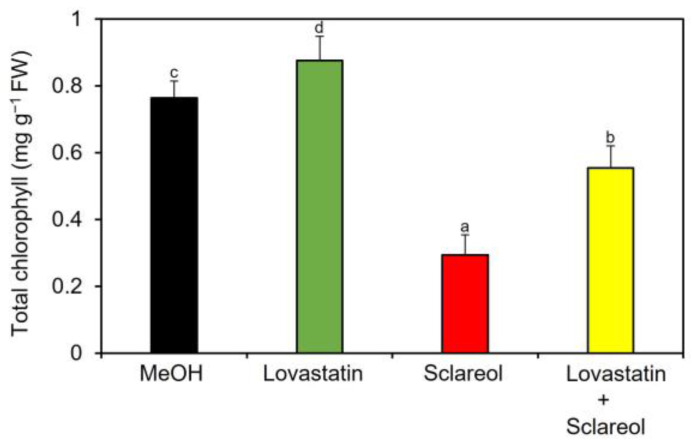
Effect of lovastatin on sclareol-induced reduction in chlorophyll. *Arabidopsis* plants were treated by soaking their roots in a solution containing 100 μM lovastatin, 100 μM sclareol, or a 50:50 mixture of lovastatin and sclareol for 48 h. The leaves of the treated plants were collected and the contents of total chlorophyll, as represented as the sum of chlorophyll *a* and b, were measured. As a control, 0.1% methanol (MeOH) was used. Values are the means ± standard deviation of three biological replicates. Different letters indicate significant differences among treatments (Tukey’s test, *p* < 0.05). The experiment was repeated three times with similar results.

## Data Availability

All the data supporting the conclusions of this study are included in the manuscript.

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
