# Peer review of "Phytosterols Are Involved in Sclareol-Induced Chlorophyll Reductions in Arabidopsis"

_plants, 2023, doi:10.3390/plants12061282_

Round 1

Reviewer 1 Report

This manuscript has a novel idea, and its results are considerable and valuable.

I would like to ask the author if they can change the graphs to be colored.

and do a fine review of the language. 

Author Response

Point 1: This manuscript has a novel idea, and its results are considerable and valuable.I would like to ask the author if they can change the graphs to be colored. and do a fine review of the language. 

Response 1: Thank you for the suggestion for organizing the graphs. We have made color graphs. The English of the manuscript has been checked and revised by a native English-speaking researcher.

Reviewer 2 Report

The authors identified internal compounds in Arabidopsis that are resposible for reducing the chlorophyll content after treatment with sclareol. To do this they treated hydroponically grown Arabidopsis plants with sclareol and fractionated it to identify fractions resposible for chlorophyll reduction. Using this approach they could find two phytosterols that could dose dependently reduce chlorophyll levels and were also ellevated upon sclareol treatment. There are a few things that are not clear to me. In the introduction the authors highlighted the beneficial effect of sclareol to act as signal to increase the disease and nematode resistance of certain plants. Yet they used the ability of sclareol to decrease chlorophyll content as marker to screen for signal transducing compounds. My concern is how is the decrease of chlorophyll associated with increased decease resistance? Plants that does not photosynthesice properly are ussually associated with decreased disease resistance. The authors should adress this discrepency in the introduction.

Secondly, the authors state that they used sclareol treated plants to identify the compounds that acted as intermediate between the sclareol signal and chlorophyll reduction. Yet they only used the sclareol treated plants to obtain different fractions that where in turn used to test for chlorophyll reduction. They then also identified phytosterols that occurs in abundance in all plant cells as these elusive signalling compounds. Can the authors please explain why they did not use any control Arabidopsis plants, not treated with sclareol, to fractionate and compare with sclareol treated plants. Why are the authors sure that the phytosterols act as intermediates in the sclareol signalling pathway and is not just a result of the reduction in chlorophyll content. 

I would also like to see more results of the fractionation experiments. Can the authors please show the data obtained for testing the different fractions against chlorophyll reduction. Whithout this data it will be almost impossible for others to repeat your experiments. It will be sufficient to add this as supplemental data.

For additional proof that phytosterols act as intermediate in the sclareol mediated reduction of chlorophyll, I recommend testing the ability of a HMGR reductase inhibitor to reduce this effect.

Author Response

Point 1: The authors identified internal compounds in Arabidopsis that are resposible for reducing the chlorophyll content after treatment with sclareol. To do this they treated hydroponically grown Arabidopsis plants with sclareol and fractionated it to identify fractions resposible for chlorophyll reduction. Using this approach they could find two phytosterols that could dose dependently reduce chlorophyll levels and were also ellevated upon sclareol treatment. There are a few things that are not clear to me. In the introduction the authors highlighted the beneficial effect of sclareol to act as signal to increase the disease and nematode resistance of certain plants. Yet they used the ability of sclareol to decrease chlorophyll content as marker to screen for signal transducing compounds. My concern is how is the decrease of chlorophyll associated with increased decease resistance? Plants that does not photosynthesice properly are ussually associated with decreased disease resistance. The authors should adress this discrepency in the introduction.

Response 1: Thank you for your suggestion. Actually, whether sclareol-induced chlorophyll reductions are related to disease resistance remains unclear. To clarify this issues was our stating point in this study. We added these descriptions to the Introduction section.

Point 2: Secondly, the authors state that they used sclareol treated plants to identify the compounds that acted as intermediate between the sclareol signal and chlorophyll reduction. Yet they only used the sclareol treated plants to obtain different fractions that where in turn used to test for chlorophyll reduction. They then also identified phytosterols that occurs in abundance in all plant cells as these elusive signalling compounds. Can the authors please explain why they did not use any control Arabidopsis plants, not treated with sclareol, to fractionate and compare with sclareol treated plants. Why are the authors sure that the phytosterols act as intermediates in the sclareol signalling pathway and is not just a result of the reduction in chlorophyll content. 

Response 2: Thank you for your suggestion. We have used sclareol-untreated Arabidopsis plants and compared with sclareol-treated plants and confirmed that chlorophyll-reducing substances exist in sclareol-treated, but not untreated, plants. We did not show such data in the original manuscript. According to the suggestion, we have shown the data as Supplementary Figure S1.   

Point 3: I would also like to see more results of the fractionation experiments. Can the authors please show the data obtained for testing the different fractions against chlorophyll reduction. Whithout this data it will be almost impossible for others to repeat your experiments. It will be sufficient to add this as supplemental data.

Response 3: Thank you for your suggestion. We have added data concerning activities of different fractions for chlorophyll reduction as Supplementary Figure S2.

Point 4: For additional proof that phytosterols act as intermediate in the sclareol mediated reduction of chlorophyll, I recommend testing the ability of a HMGR reductase inhibitor to reduce this effect.

Response 4: Thank you for your suggestion. We used lovastatin, an inhibitor of HMGR, to address whether phytosterols act as intermediate in the sclareol-mediated reduction of chlorophyll. We have added new data as Figure 6.

Reviewer 3 Report

Authors must answer the following questions:

-What aspects of growth and development are you referring to? (Lines 83 and 84)

-Explain the reason for choosing 4-week-old plants. (Line 161)
-
The procedure applied to the chemical treatment was proposed by which author?

Author Response

Authors must answer the following questions:

Point 1: -What aspects of growth and development are you referring to? (Lines 83 and 84)

Response 1: Thank you for your suggestion. We have added a detailed explanation.

Point 2: -Explain the reason for choosing 4-week-old plants. (Line 161)

Response 2: Because 4-week-old plants exhibited an increase in chlorophyll reduction compared to 8-week-old plants, we used 4-week-old plants. This explanation has been added to the first paragraph in the Results and Discussion section.

Point 3: -The procedure applied to the chemical treatment was proposed by which author?

Response 3: The procedure for the chemical treatment was proceeded according to our previous works. This explanation has been added to the Materials and Methods section.

Round 2

Reviewer 2 Report

All my concerns were adressed in full. I recommend acceptance for publication. I found the manuscript very interesting and hope the topic will be studied further in the future.